# Mathematical modeling of the transmission of SARS-CoV-2—Evaluating the impact of isolation in São Paulo State (Brazil) and lockdown in Spain associated with protective measures on the epidemic of CoViD-19

**Hyun Mo Yang**[1]*, **Luis Pedro Lombardi Junior**[1], **Fábio Fernandes Morato Castro**[2], **Ariana Campos Yang**[2]

**1** Department of Applied Mathematics, State University of Campinas, Campinas, São Paulo, Brazil, **2** Division of Allergy and Immunology, General Hospital of the Medicine School of University of São Paulo, São Paulo, São Paulo, Brazil

* hyunyang@ime.unicamp.br

**Data Availability Statement:** The data that support the findings of this study are openly available: São

## Abstract

Coronavirus disease 2019 (CoViD-19), with the fatality rate in elder (60 years old or more) being much higher than young (60 years old or less) patients, was declared a pandemic by the World Health Organization on March 11, 2020. A mathematical model considering young and elder subpopulations under different fatality rates was formulated based on the natural history of CoViD-19 to study the transmission of the severe acute respiratory syndrome coronavirus 2 (SARS-CoV-2). The model considered susceptible, exposed, asymptomatic, pre-symptomatic, mild CoViD-19, severe CoViD-19, and recovered compartments, besides compartments of isolated individuals and those who were caught by test. This model was applied to study the epidemiological scenario resulting from the adoption of quarantine (isolation or lockdown) in many countries to control the rapid propagation of CoViD-19. We chose as examples the isolation adopted in São Paulo State (Brazil) in the early phase but not at the beginning of the epidemic, and the lockdown implemented in Spain when the number of severe CoViD-19 cases was increasing rapidly. Based on the data collected from São Paulo State and Spain, the model parameters were evaluated, and we obtained a higher estimation for the basic reproduction number $R_0$ (9.24 for São Paulo State, and 8 for Spain) compared to the currently accepted estimation of $R_0$ around 2 using the SEIR (susceptible, exposed, infectious, and recovered compartments) model. In comparison with the lockdown in Spain, the relatively early adoption of the isolation in São Paulo State resulted in enlarging the period of the first wave of the epidemic and delaying its peak. The model allowed to explain the flattening of the epidemic curves by quarantine when associated with the protective measures (face mask, washing hands with alcohol and gel, and social distancing) adopted by the population. The description of the epidemic under quarantine and protections can be a background to foreseen the

Paulo State at https://www.saopaulo.sp.gov.br/coronavirus#numero-vacinacao and https://www.seade.gov.br; and Spain at https://cnecovid.isciii.es/covid19/ and https://www.ine.es/en/index.htm. Other public database: https://www.populationpyramid.net/.

**Funding:** The author(s) received no specific funding for this work.

**Competing interests:** The authors have declared that no competing interests exist.

epidemiological scenarios from the release strategies, which can help guide public health policies by decision-makers.

## Introduction

Coronavirus disease 2019 (CoViD-19) is caused by severe acute respiratory syndrome corona-virus 2 (SARS-CoV-2), a strain of the RNA-based SARS-CoV-1. The SARS-CoV-2 (new coronavirus) can be transmitted by droplets that escape the lungs through coughing or sneezing and infect humans (direct transmission), or they are deposited in surfaces and infect humans when in contact with this contaminated surface (indirect transmission) [1, 2]. These transmission routes resulted in a rapid spreading of this virus, and the World Health Organization (WHO) declared CoViD-19 a pandemic on March 11, 2020. In general, the fatality rate in elder patients (60 years or more) is much higher than those with 60 years or less [3].

Currently, there is not a vaccine neither an effective treatment. Hence, at the beginning of CoViD-19 outbreaks, quarantine was the primary way of controlling the dissemination of the new coronavirus in a population [4]. However, there is evidence that individual (protection of mouth and nose using face mask and protection of eyes, and washing hands with alcohol and gel) and collective (social distancing) protective measures diminish the transmission of CoViD-19 [5]. The decrease in the incidence of CoViD-19 by quarantine, known as flattening the curve of an epidemic, can be quantified by mathematical modeling.

Initially, computational models (especially, agent-based model) to describe the influenza epidemic were adapted and applied to estimate the spreading of SARS-CoV-2. Koo *et al.* [6] used such a model to study the propagation of the new coronavirus in Singapore, assuming that the basic reproduction number, denoted by $R_0$, was around 2. The same approach was made by Ferguson *et al.* [7] to investigate the impact of non-pharmaceutical interventions (isolation of susceptible persons) named mitigation and suppression. Briefly, mitigation reduces the basic reproduction number $R_0$ but not lower than one, while suppression reduces the basic reproduction number lower than one. Their model was simulated assuming $R_0$ around 2.5, and predicted the numbers of severe cases and deaths due to CoViD-19 without interventions and compared them with those numbers when implemented quarantine (mitigation or suppression) in a population. However, instead of assuming a specific value for $R_0$, Li *et al.* [8] performed a stochastic simulation of SEIR (susceptible, exposed, infectious, and recovered compartments) model incorporating the rapid dissemination of new coronavirus due to undocumented infections. They estimated the effective reproduction number $R_{ef}$ around 2.4. In the SEIR model, the severe CoViD-19 cases were used to estimate the basic reproduction number $R_0$; however, those individuals are isolated in hospitals receiving treatment [9].

Mathematical models based on a well-documented natural history of the disease allow us to understand the progression of viral infection and provide mathematical expression to estimate $R_0$, which is related to the magnitude of efforts to eradicate an infection [10]. When a simple SIR model is considered to describe the CoViD-19 epidemic, it is expected to estimate $R_0$ around 2. Fortunately, the knowledge about the natural history of CoViD-19 is being improved every day as the epidemic evolves. Consequently, incorporating novel aspects of this epidemic can benefit mathematical modeling. In Yang *et al.* [11], a mathematical model encompassing two subpopulations based on the different fatality rates in young (60 years old or less) and elder (60 years old or more) subpopulations was developed aiming to study the impacts of the quarantine and further relaxation (release) on the epidemic of CoViD-19. That

model was applied to São Paulo State (Brazil) to describe the epidemiological scenario considering intermittent pulses in isolation and release. The isolation of the susceptible persons (non-pharmaceutical interventions [7]) jumps down $R_{ef}$ as the herd immunity jumps it down by the vaccination [10].

We improved the model presented in [11], allowing the transmission of the infection by persons manifesting mild CoViD-19 symptoms and incorporating the protective measures that reduced the virus's transmission. Briefly, the model considers susceptible, exposed, asymptomatic, pre-symptomatic, mild CoViD-19, severe CoViD-19, and recovered compartments based on the natural history of CoViD-19. The general model also considers the compartments of isolated persons and those who were caught by test. The protection conferred to susceptible persons by quarantine and by adopting protective measures by the non-isolated (circulating) subpopulation is named "herd protection". The model proposed here is applied to evaluate the impacts of herd protection on the epidemic in São Paulo State and Spain.

The widespread of CoViD-19 in Spain led to adopting a rigid quarantine (lockdown, hereafter), which is an extreme measure to control an epidemic's quick increase [12, 13]. São Paulo State, however, implemented a partial quarantine (isolation, hereafter) in the population to avoid critical epidemiological scenarios that occurred in Spain [14] and Italy [15]. Based on the data collection of severe CoViD-19 cases and deaths from São Paulo State and Spain, we aim to estimate the model parameters, the proportion of the population in isolation/lockdown, and the reduction in the transmission rates by adopting the protective measures. The estimated parameters allow us to calculate the basic reproduction number $R_0$ for São Paulo State and Spain and compare the CoViD-19 epidemiological scenarios yielded in both regions.

## Materials and methods

We present a general model to describe the CoViD-19 epidemic considering the quarantine and relaxation. However, to characterize the epidemic restricted to the quarantine, the system of Eqs (A.2-A.4) in S1 Appendix is reduced by dropping out the relaxation ($l_{ij} = 0$) as well as the isolation due to mass test ($\eta_j = \eta_{1j} = \eta_{2j} = 0$), treatment ($\theta_j = 0$), and educational campaign ($\varpi_j = \xi_j = 0$), with $j = y$, $o$. The description of the model variables and parameters are given in Section A.1 in S1 Appendix. In Table 1, we summarize the model classes (or variables) of the reduced model.

In Table 2, we summarize the reduced model parameters. The description of the assigned values (for elder classes, values are between parentheses) can be found in Section D.2 in S1 Appendix. The transmission rates $\beta_{1j}$, $\beta_{2j}$, and $\beta_{3j}$, additional mortality rate $\alpha_j$, the proportion in isolation $k_j$, with $j = y$, $o$, and the protection and reduction factors $\varepsilon$ and $\omega$ are estimated in the next section.

**Table 1. Summary of the model variables ($j = y$, $o$).**

| Symbol | Meaning |
| --- | --- |
| $S_j$ | Susceptible persons |
| $Q_j$ | Isolated among susceptible persons |
| $E_j$ | Exposed and incubating new coronavirus persons |
| $A_j$ | Asymptomatic persons |
| $D_{1j}$ | Pre-diseased (pre-symptomatic) persons |
| $Q_{2j}$ | Mild (non-hospitalized) CoViD-19 persons |
| $D_{2j}$ | Severe (hospitalized) CoViD-19 persons |
| $I$ | Immune (recovered) persons |

**Table 2. Summary of the reduced model parameters ($j = y, o$) and values (rates in *days*$^{-1}$, time in *days* and proportions are dimensionless).** Some values are calculated (#) or assumed ($) or obtained from the liteature (*) or estimated (**). The values (&) correspond to São Paulo State. For Spain, $\phi = \mu = 1/(83.4 \times 365)$ *days*$^{-1}$, $\varphi = 1.14 \times 10^{-5}$ *days*$^{-1}$, and $\tau^{is}$ is March 16.

| Symbol | Meaning | Value |
|---|---|---|
| $\mu$ | Natural mortality rate | $1/(78.4 \times 365)$*& |
| $\phi$ | Birth rate | $1/(78.4 \times 365)$*& |
| $\varphi$ | Aging rate | $6.7 \times 10^{-6}$#& |
| $\sigma_y(\sigma_o)$ | Incubation rate | $1/5.8(1/5.8)$# |
| $\gamma_y(\gamma_o)$ | Recovery rate of asymptomatic persons | $1/12(1/14)$* |
| $\gamma_{1y}(\gamma_{1o})$ | Infection rate of pre-diseased persons | $1/4(1/4)$* |
| $\gamma_{2y}(\gamma_{2o})$ | Recovery rate of severe CoViD-19 | $1/12(1/21)$* |
| $\gamma_{3y}(\gamma_{3o})$ | Infection rate of mild CoViD-19 persons | $1/13(1/16)$* |
| $\tau^{is}$ | Time of the introduction of isolation | March 24*& |
| $z_y(z_o)$ | Proportion of transmission by mild CoViD-19 persons | $0.5(0.2)$\$ |
| $\chi_y(\chi_o)$ | Proportion of remaining as asymptomatic persons | $0.98(0.95)$\$ |
| $p_y(p_o)$ | Proportion of asymptomatic persons | $0.8(0.8)$# |
| $m_y(m_o)$ | Proportion of mild (non-hospitalized) CoViD-19 | $0.92(0.75)$# |
| $\varepsilon$ | Protection factor | ** |
| $\omega$ | Reduction factor | ** |
| $k_y(k_o)$ | Proportion in isolation | ** |
| $\alpha_y(\alpha_o)$ | Additional mortality rate | ** |
| $\beta_{1y}(\beta_{1o})$ | Transmission rate due to asymptomatic persons | ** |
| $\beta_{2y}(\beta_{2o})$ | Transmission rate due to pre-diseased persons | ** |
| $\beta_{3y}(\beta_{3o})$ | Transmission rate due to mild CoViD-19 persons | ** |

Therefore, the reduced model has the equations for susceptible individuals

$$\begin{cases} \dfrac{d}{dt}S_y = \phi N - (\varphi + \mu)S_y - \lambda S_y - k_y S_y \delta(t - \tau_y^{is}) \\[2mm] \dfrac{d}{dt}S_o = \varphi S_y - \mu S_o - \lambda \psi S_o - k_o S_o \delta(t - \tau_o^{is}), \end{cases} \tag{1}$$

for isolated and infectious individuals, with $j = y, o$,

$$\begin{cases} \dfrac{d}{dt}Q_{1j} = k_j S_j \delta(t - \tau_j^{is}) - \mu Q_j \\[2mm] \dfrac{d}{dt}E_j = \lambda(\delta_{jy} + \psi \delta_{jo})S_j - (\sigma_j + \mu)E_j \\[2mm] \dfrac{d}{dt}A_j = l_j \sigma_j E_j - (\gamma_j + \mu)A_j \\[2mm] \dfrac{d}{dt}D_{1j} = (1 - p_j)\sigma_j E_j - (\gamma_{1j} + \mu)D_{1j} \\[2mm] \dfrac{d}{dt}Q_{2j} = (1 - \chi_j)\gamma_j A_j + m_j \gamma_{1j} D_{1j} - (\gamma_{3j} + \mu)Q_{2j} \\[2mm] \dfrac{d}{dt}D_{2j} = (1 - k_j)\gamma_{1j} D_{1j} - (\gamma_{2j} + \mu + \alpha_j)D_{2j}, \end{cases} \tag{2}$$

and for recovered individuals,

$$\frac{d}{dt}I = \chi_y\gamma_y A_y + \gamma_{3y}D_{1y} + \gamma_{2y}D_{2y} + \chi_o\gamma_o A_o + \gamma_{3o}D_{1o} + \gamma_{2o}D_{2o} - \mu I, \tag{3}$$

where $N_j = S_j + E_j + A_j + D_{1j} + Q_{2j} + D_{2j}$, and $N = N_y + N_o + I$ obeys Eq (A.5) in S1 Appendix. The force of infection $\lambda$ is given by Eq (A.1) in S1 Appendix.

The system of non-autonomous and non-linear differential Eqs (1–3) is simulated permitting pulse intervention to the boundary conditions. Hence, the equations for susceptible and isolated persons become

$$
\begin{cases}
\dfrac{d}{dt}S_y &=& \phi N - (\varphi + \mu)S_y - \lambda S_y \\[2mm]
\dfrac{d}{dt}S_o &=& \varphi S_y - \mu S_o - \lambda\psi S_o \\[2mm]
\dfrac{d}{dt}Q_j &=& -\mu Q_j,
\end{cases}
\tag{4}
$$

$j = y, o$, and other equations are the same. Hence, for the system of Eqs (1–3), the initial conditions (at $t = 0$) are, for $j = y, o$,

$$S_j(0) = N_{0j}, \quad X_j(0) = n_{X_j}, \quad \text{where} \quad X_j = Q_j, E_j, A_j, D_{1j}, Q_{2j}, D_{2j}, I, \tag{5}$$

where $N_{0y}$ and $N_{0o}$ are the size of young and elder subpopulations, with $N(0) = N_0 = N_{0y} + N_{0o}$, and $n_{X_j}$ is a non-negative number. For instance, $n_{E_y} = n_{E_o} = 0$ means that there is not any exposed person (young and elder) at the beginning of the epidemic.

The boundary conditions describing the quarantine implemented at $t = \tau^{is}$ are

$$S_j(\tau^{is+}) = S_j(\tau^{is-})(1 - k_j) \quad \text{and} \quad Q_j(\tau^{is+}) = Q_j(\tau^{is-}) + S_j(\tau^{is-})k_j, \tag{6}$$

plus

$$X_j(\tau^{is+}) = X_j(\tau^{is-}), \quad \text{where} \quad X_j = E_j, A_j, D_{1j}, Q_{2j}, D_{2j}, I, \tag{7}$$

where we have $\tau^{is-} = \lim_{t \to \tau^{is}} t$ (for $t < \tau^{is}$), and $\tau^{is+} = \lim_{\tau^{is} \leftarrow t} t$ (for $t > \tau^{is}$). If quarantine is applied to a completely susceptible population at $t = 0$, there are not any infectious persons, so $S(0) = N_0$. If quarantine is done at $t = \tau_j^{is}$ without a screening of persons harboring the virus, then many of the asymptomatic persons could be isolated with susceptible persons.

The epidemiological scenario of quarantine is obtained by the solution of the system of Eqs (1–3). The simulation of this system provides the epidemic curve (severe CoViD-19 cases $D_2$), and the numbers of susceptible ($S$) and recovered ($I$) persons. However, the following epidemiological parameters (the initial conditions (5) supplied to the system of equations determine all initial conditions below) are derived:

**(1)** The number of non-isolated (circulating) persons $S_j$ is obtained from Eq (4), and the number of circulating plus isolated susceptible persons $S^{tot}$ is obtained by

$$S^{tot} = S_y^{tot} + S_o^{tot}, \qquad \text{with} \quad
\begin{cases}
S_y^{tot} = S_y + Q_y \\[2mm]
S_o^{tot} = S_o + Q_o,
\end{cases}
\tag{8}$$

where $S_y^{tot}$ and $S_o^{tot}$ are the numbers of susceptible, respectively, young and elder persons.

(2) The numbers of new cases of CoViD-19 $\Phi_y$ and $\Phi_y$ are

$$
\begin{cases}
\dfrac{d}{dt}\Phi_y = \lambda S_y \\[2mm]
\dfrac{d}{dt}\Phi_o = \lambda\psi S_o,
\end{cases}
\quad \text{with} \quad \Phi = \Phi_y + \Phi_o,
\tag{9}
$$

where $\Phi_y(0) = E_y(0)$ and $\Phi_o(0) = E_o(0)$.

(3) The numbers of accumulated severe CoViD-19 cases $\Omega_y$ and $\Omega_o$ are given by the exits from $D_{1y}$, $Q_{1y}$, $D_{1o}$, and $Q_{1o}$, and entering into classes $D_{2y}$ and $D_{2o}$, that is,

$$
\begin{cases}
\dfrac{d}{dt}\Omega_y = \left(1 - m_y\right)\gamma_{1y}\left(D_{1y} + Q_{1y}\right) \\[2mm]
\dfrac{d}{dt}\Omega_o = (1 - m_o)\gamma_{1o}(D_{1o} + Q_{1o}),
\end{cases}
\quad \text{with} \quad \Omega = \Omega_y + \Omega_o,
\tag{10}
$$

with $\Omega_y(0) = \Omega_{y0}$ and $\Omega_o(0) = \Omega_{o0}$. The daily severe CoViD-19 cases $\Omega_d$ is, considering $\Delta t = t_i - t_{i-1} = \Delta t = 1\ day$,

$$
\Omega_d(t_i) = \int_{t_{i-1}}^{t_i} \frac{d}{dt}\Omega dt = \Omega(t_i) - \Omega(t_{i-1}),
\tag{11}
$$

where $\Omega_d(0) = \Omega_{d0}$ is the first observed CoViD-19 case at $t_0 = 0$, with $i = 1, 2, \cdots$, and $t_1 = 1$ is the next day in the calendar time, and so on.

(4) The number of deaths due to severe CoViD-19 cases is

$$
\Pi = \Pi_y + \Pi_o, \quad \text{where} \quad
\begin{cases}
\dfrac{d}{dt}\Pi_y = \alpha_y D_{2y}, \quad \text{with} \quad \Pi_y(0) = 0 \\[2mm]
\dfrac{d}{dt}\Pi_o = \alpha_o D_{2o}, \quad \text{with} \quad \Pi_o(0) = 0.
\end{cases}
\tag{12}
$$

In the estimation of the additional mortality rates, we must bear in mind that the registration times of the new cases and deaths do not have direct correspondence, somewhat they are delayed by $\Delta$ days, that is, $\Pi_y(t + \Delta) = \alpha_y D_{2y}(t)$, for instance. We can estimate the severity case fatality rate as the quotient $\Pi/\Omega$, and the infection fatality rate as $\Pi/\Phi$.

The model parameters are estimated using the registered data from São Paulo State (February 26 to May 7, 2020) and Spain (January 31 to May 20, 2020). We calculate the basic ($R_0$) and effective ($R_{ef}$) reproduction numbers obtained from the analysis of the steady-state corresponding to the system of Eqs (A.2-A.4) in S1 Appendix, which are found in Section A.2 in S1 Appendix. We perform the sensitivity analysis of $R_0$, which is given in Section A.3 in S1 Appendix.

## Results

In the preceding section, we presented a mathematical model to describe the transmission of SARS-CoV-2, which is applied to describe the epidemiological scenarios of isolation in São Paulo State and lockdown in Spain. We obtained the basic reproduction number $R_0$ and the effective reproduction number $R_{ef}$ (see Section A.2 in S1 Appendix).

In Section B in S1 Appendix, the accumulated CoViD-19 data follow three distinct trends. These periods in São Paulo State describe the epidemic occurring naturally, with isolation, and isolation associated with protective measures. In Spain, the three periods represent the natural epidemic, the epidemic during the transition from natural to lockdown, and the epidemic occurring in lockdown. Considering these three trends shown by the CoViD-19 data, we evaluate the model parameters (see Section C in S1 Appendix).

It is worth stressing that the first period of data corresponding to the natural epidemic of CoViD-19 in São Paulo State and Spain is a unique opportunity to estimate $R_0$. It is in concordance with the definition of the basic reproduction number: Completely susceptible population without constraints (interventions) [10]. Based on this estimation, all subsequent non-pharmaceutical interventions (herd protection) applied to flattening the epidemic curve can be assessed. In the preceding section, we described the quarantine as a pulse, resulting in a sharp fall (jump down) in the effective reproduction number $R_{ef}$, as observed in pulse vaccination [10]. We estimate the magnitude of the jump down in $R_{ef}$ due to the quarantine and protective measures.

## CoViD-19 in São Paulo State—Isolation

São Paulo State has 44.6 million inhabitants with 15.3% of elder population (60 years old or more) [16], and the demographic density is $177/km^2$ [17]. The first confirmed case of CoViD-19 occurred on February 26, the first death due to CoViD-19 on March 16, and on March 24, São Paulo State implemented the isolation of people in non-essential activities.

In Section C.1 in S1 Appendix, we evaluate the model parameters based on the daily collected data (see B.1 and B.2 Figs in Section B.1 of S1 Appendix), using the estimation method described in Section D.1 in S1 Appendix. We summarize the estimated values using data from February 26 to May 7 (see C.1-C.4 Figs in Section C.1 of S1 Appendix):

1. Data from February 26 to April 3—The transmission rates $\beta_y = 0.78$ and $\beta_o = 0.90$ (both in $days^{-1}$), giving $R_0 = 9.24$; the additional mortality rates $\alpha_y = 0.00185$ and $\alpha_o = 0.0071$ (both in $days^{-1}$).

2. Data from March 24 to April 12—The proportion in isolation of susceptible persons $k = 0.53$.

3. Data from April 4 to May 7—The protective factor $\varepsilon = 0.5$ reducing the transmission rates to $\beta'_y = 0.39$ and $\beta'_o = 0.45$ (both in $days^{-1}$).

Using these values, we describe the epidemiological scenario of isolation associated with protective measures.

In Fig 1, we show the effects of interventions on the dynamic of the new coronavirus. As interventions are added (isolation followed by protective measures), we observe decreasing in the peaks of severe CoViD-19 $D_2$, which move to the right. Fig 1(a) shows the natural epidemic, epidemic considering only isolation, and epidemic occurring with isolation associated with protective measures. In Fig 1(b), we show the number of immune (recovered) persons $I$ corresponding to the three cases shown in Fig 1(a). The curves of $I$ have a sigmoid shape.

On June 15, the date proposed to initiate the relaxation of quarantine, in the absence of interventions ($k = 0$ and $\varepsilon = 1$), the numbers of immune persons $I_y$, $I_o$, and $I$ increase from zero to, respectively, 36.92 million, 6.505 million, and 43.43 million. When interventions (isolation and protective measures) are adopted, the numbers are 6.12 million (16.6%), 1.14 million (16.5%), and 7.26 million (16.7%) on June 15. The percentage between parentheses is the ratio between with and without interventions $I(k, \varepsilon)/I(0, 1)$.

(a)

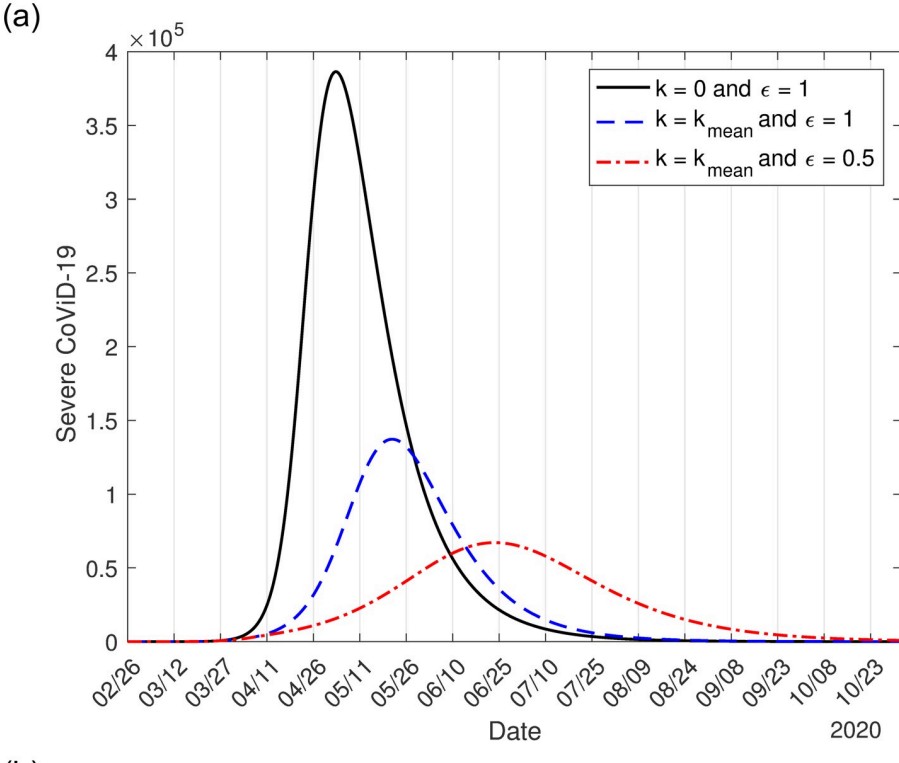

(b)

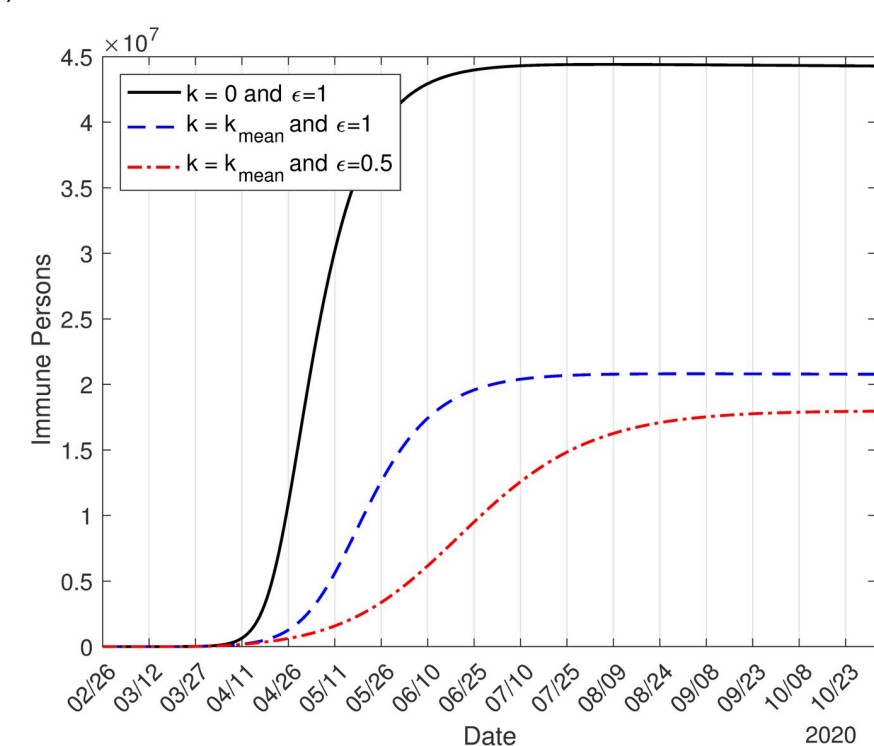

**Fig 1.** The curves of the natural epidemic ($k = 0$ and $\varepsilon = 1$), epidemic considering only isolation ($k = 0.53$ and $\varepsilon = 1$), and epidemic occurring with isolation and protective measures ($k = 0.53$ and $\varepsilon = 0.6$) (a), and the number of immune persons $I$ (b).

**Table 3. The peaks for young, elder, and total persons in the natural epidemic, isolation alone, and isolation associated with protective measures.** The percentage between parentheses is the ratio between with and without interventions $D_2(k, \varepsilon)/D_2(0,1)$ and the peak's occurrence date.

|  | young ($y$) | elder ($o$) | all persons |
|---|---|---|---|
| Natural epidemic ($k = 0$, $\varepsilon = 1$) | 224, 200 (May 2) | 162, 200 (May 4) | 386, 400 (May 3) |
| Isolation only ($k = 0.53$, $\varepsilon = 1$) | 77, 320 (34%) (May 21) | 60, 020 (37%) (May 22) | 137, 200 (36%) (May 21) |
| Isolation and protection ($k = 0.53$, $\varepsilon = 0.5$) | 36, 010 (16%) (June 22) | 31, 160 (19%) (June 24) | 67, 140 (18%) (June 23) |

In Table 3, we show the peaks of $D_2$. Compared with the natural epidemic, the peaks of the flattened epidemic curves decrease around to 35% and 18% in isolation alone and isolation associated with protective measures. The protective measures yielded an additional decrease of 17%. Another benefit of the protective measures is a further delay in one month to occur the peak.

Due to isolation and protective measures, many people remain as susceptible. In Fig 2 we show the circulating susceptible persons $S_y$, $S_o$ and $S = S_y + S_o$ (a), and circulating plus isolated susceptible persons $S_y^{tot}$, $S_o^{tot}$ and $S^{tot} = S_y^{tot} + S_o^{tot}$ (b), using Eq (8). Remember that $S_y^{tot}$ differs from $S_y$ just after the introduction of isolation (March 24).

In Table 4, we show the numbers of susceptible persons $S_y$, $S_o$, and $S = S_y + S_o$ without any interventions ($k = 0$ and $\varepsilon = 1$), with interventions (isolation and protective measures), and adding isolated persons $S_y^{tot}$, $S_o^{tot}$ and $S^{tot} = S_y^{tot} + S_o^{tot}$ at the end of the first wave of the epidemic. Observe that, on June 15, the sum of the susceptible persons in circulation and those in isolation is such that there are more than 750 times and 23, 000 times, respectively, susceptible young and elder persons in comparison with epidemic without any intervention. Hence, if all persons are released without planning, the second wave will be intense, infecting much more elder persons. In the absence of vaccine and effective treatment, interventions to reduce the transmission of SARS-CoV-2 must be continued for a long time to avoid the rebounding of the epidemic or a second wave.

It is essential to estimate the basic reproduction number, which portrays the beginning and ending phases of an epidemic [18]. During the epidemic, however, the effective reproduction number determines the risk of transmission of the infection. We use the approximate effective reproduction number $R_{ef}$, given by Eq (A.18) in S1 Appendix, to follow the intensity of the epidemic, remembering that $R_{ef} > 1$ implies epidemic in expansion, while $R_{ef} < 1$, in contraction. Fig 3 illustrates the effective reproduction number $R_{ef}$ and $D_2$ during the epidemic, with (a) and without (b) interventions. To be fitted together in the same frame with $R_{ef}$, the curve of $D_2$ was divided by 7, 000 (a) and 40, 000 (b). The curve of $R_{ef}$ follows the shape of susceptible persons, as shown in Fig 2, as expected. At the peak of the epidemic, the effective reproduction number is lower than one; hence we have $R_{ef} = 1$ occurring on June 14 (a) and April 6 (b).

As the epidemic evolves, the effective reproduction number varies, as shown in Fig 3(a). At the beginning of the epidemic, on February 26, we have $R_{ef} = R_0 = 9.24$, on March 24, a jump down occurred to $R_{ef} = 4.35$ due to the isolation, and a new jump down occurs to $R_{ef} = 2.15$ on April 4 when protective measures were adopted. On June 15, when the release will begin, we have $R_{ef} = 0.98$, but in the ascending phase of the epidemic. The knowledge of $R_{ef}$ may help public health authorities plan the release strategies.

We used the accumulated data shown in B.1(b) Fig in S1 Appendix and $\Omega$ given by Eq (10) to estimate the transmission rates $\beta_y$ and $\beta_o$, the proportion in isolation $k$, and the protective factor $\varepsilon$. The curve labeled $\varepsilon = 0.5$ in C.3(b) Fig in S1 Appendix is the estimated curve $\Omega$, from

(a)

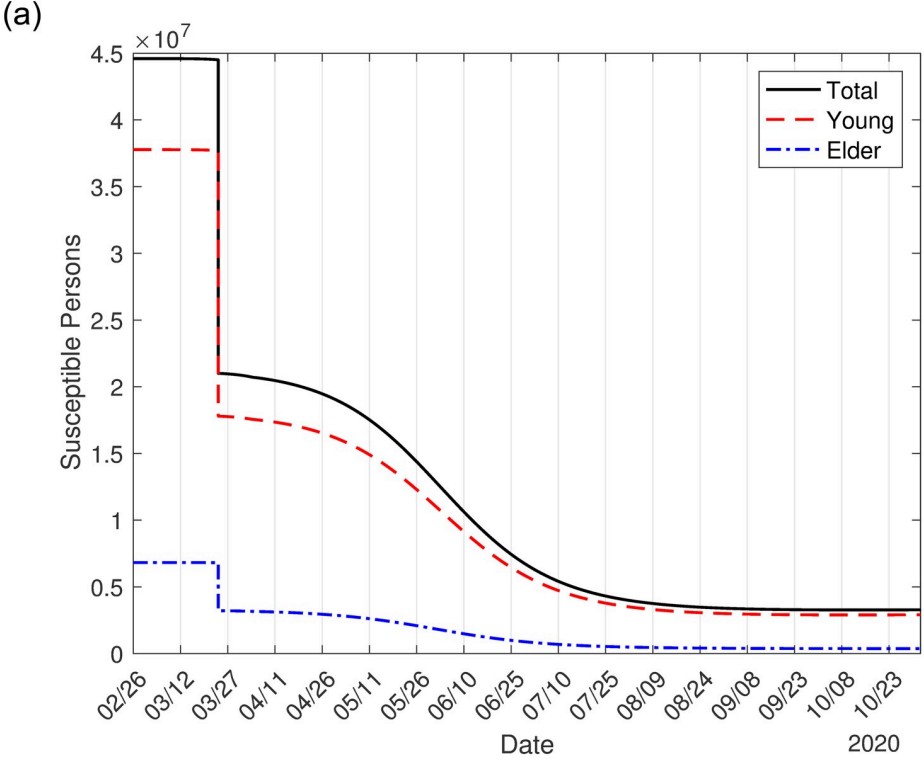

(b)

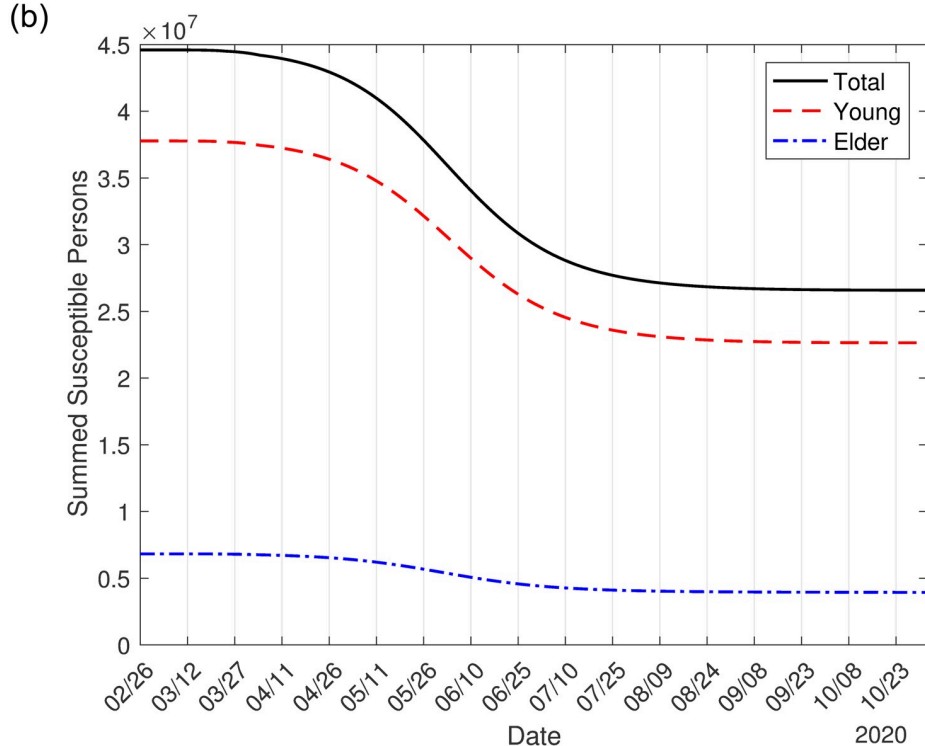

**Fig 2.** The circulating susceptible young $S_y$, elder $S_o$ and total $S = S_y + S_o$ persons (a), and the sum of the circulating and isolated susceptible populations $S_y^{tot}$, $S_o^{tot}$ and $S^{tot} = S_y^{tot} + S_o^{tot}$ (b).

**Table 4. The numbers of susceptible persons $S_y$, $S_o$, and $S = S_y + S_o$ without and with interventions, and $S_y^{tot}$, $S_o^{tot}$ and $S^{tot} = S_y^{tot} + S_o^{tot}$ at the end of the first wave of the epidemic.** The percentage between parentheses is the ratio between with and without interventions $S(k, \varepsilon)/S(0, 1)$.

| | young ($S_y$) | elder ($S_o$) | all persons ($S_y + S_o$) |
|---|---|---|---|
| Natural epidemic | 37, 000 | 210 | 37, 210 |
| Isolation and protection (circulating) | 8.15 million (22, 027%) | 1.3 million (619, 047%) | 9.5 million (25, 531%) |
| Isolation and protection (adding isolated) | 28 million (75, 676%) | 4.9 million (2, 333, 333%) | 33 million (88, 686%) |

which the curve of severe cases $D_2$ was derived, corresponding to the most flattened curve shown in Fig 1(a). Now, from the estimated curve of $\Omega$, we derive the daily cases $\Omega_d$ given by Eq (11). In Fig 4(a), we show the calculated curve $\Omega_d$ and daily cases presented in B.1(a) Fig in S1 Appendix. In Fig 4(b), we show the initial part of the estimated curve $\Omega$ with observed data $\Omega^{ob}$, the extended $\Omega_d$ and daily observed cases $\Omega_d^{ob}$, and severe cases $D_2$. The peaks of $\Omega_d$ and $D_2$ occur, respectively, on June 12 and 23.

On June 12, the peak of the daily cases of CoViD-19 predicted by the model reaches 5, 286. On June 23, the peak of $D_2$ estimated by the model is 67, 140, and the number of accumulated cases $\Omega$ is 243, 000, which is 362% of the peak of $D_2$, and 63% of cases when the first wave of epidemic ends (386, 700). These values provided by the model correspond to the epidemic with isolation without release.

## CoViD-19 in Spain—Lockdown

Spain has 47.4 million inhabitants [19] with 25.8% of elder population [20], and the demographic density is 92.3/$km^2$ [17]. In Spain, the first confirmed case of CoViD-19 occurred on January 31, 2020. However, the daily registering of CoViD-19 began on February 20 (3 cases), the first 28 deaths were registered on March 8 when reached 1, 535 cases, and on March 16, the lockdown was implemented.

In Section C.2 in S1 Appendix, we evaluate the model parameters based on the daily collected data (see B.3 Fig in Section B.2 of S1 Appendix), using the estimation method described in Section D.1 in S1 Appendix. We summarize the estimated values using data from January 31 to May 20 (see C.5-C.8 Figs in Section C.8 of S1 Appendix):

1. Data from January 31 to March 21—In the natural epidemic, the estimated values are $\beta_y$ = 0.67 and $\beta_o$ = 0.74 (both in $days^{-1}$) for the transmission rates, giving $R_0$ = 8.0, and the additional mortality rates are $\alpha_y$ = 0.00273 and $\alpha_o$ = 0.0105 (both in $days^{-1}$).

2. Data from March 22 to 28—During the epidemic in transition, for the proportion in the lockdown of susceptible person $k$ = 0.9, the estimated transmission rates are $\beta'_y = 0.45$ and $\beta'_o = 0.49$ (both in $days^{-1}$) in the isolated population.

3. Data from March 24 to May 20—In the epidemic during the lockdown, for the protective factor $\varepsilon$ = 0.5, the estimated transmission rates are $\beta'_y = 0.34$ and $\beta'_o = 0.391$ (both in $days^{-1}$) in the circulating population, while in the people in lockdown, for the decreasing factor $\omega$ = 11.5, $\beta'_y = 0.059$ and $\beta'_o = 0.068$ (both in $days^{-1}$).

Using these values, we describe the epidemiological scenario of lockdown associated with protective measures.

In Fig 5, we show the effects of interventions on the dynamic of the new coronavirus. During the three phases of the epidemic (natural, in transition, and effective lockdown), we

(a)

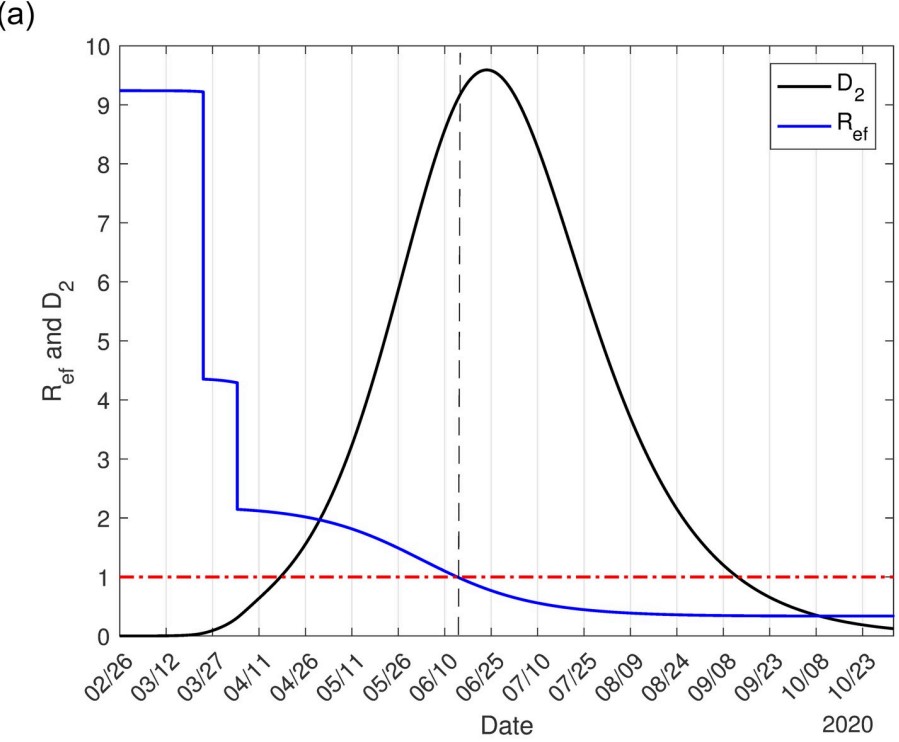

(b)

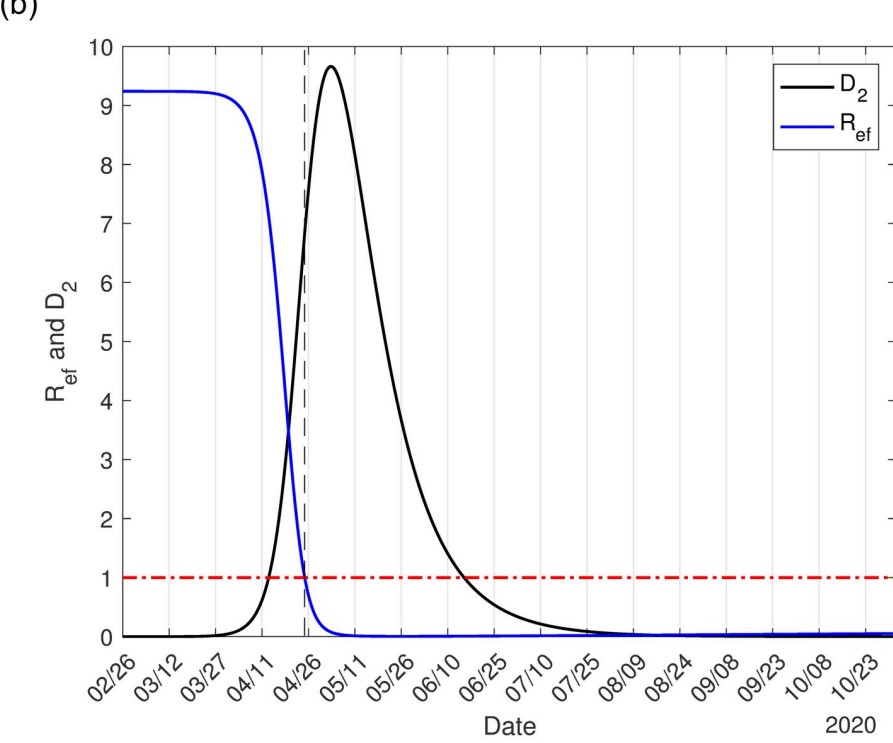

**Fig 3.** The effective reproduction number $R_{ef}$ for epidemic with isolation and protective measures (a), and natural epidemic (b) in São Paulo State. The number of severe covid-19 cases $D_2$ must be multiplied by 7, 000 (a) and 40, 000 (b).

(a)

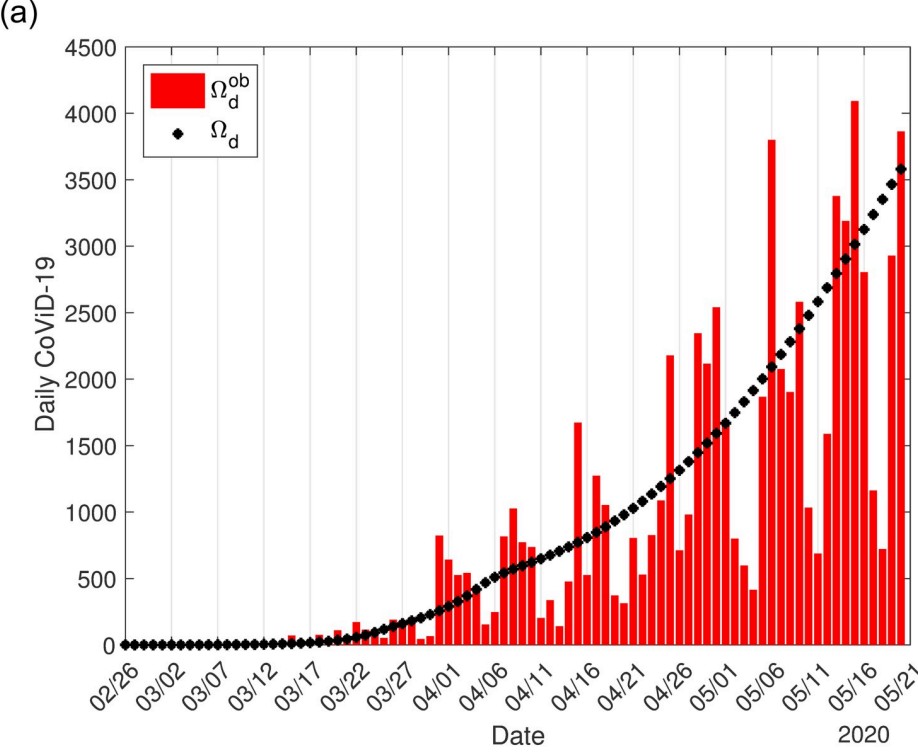

(b)

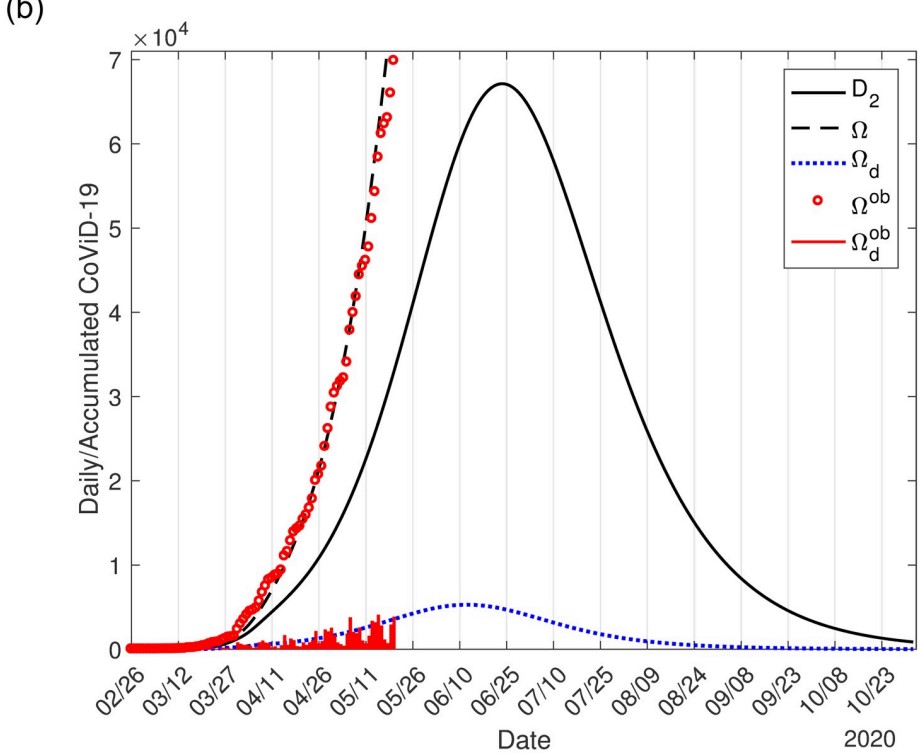

**Fig 4.** The calculated curve $\Omega_d$ and observed daily cases in São Paulo State (a), and the initial part of the estimated curve $\Omega$ with observed data $\Omega^{ob}$, the extended $\Omega_d$ and daily observed cases $\Omega_d^{ob}$, and severe covid-19 cases $D_2$ (b).

(a)

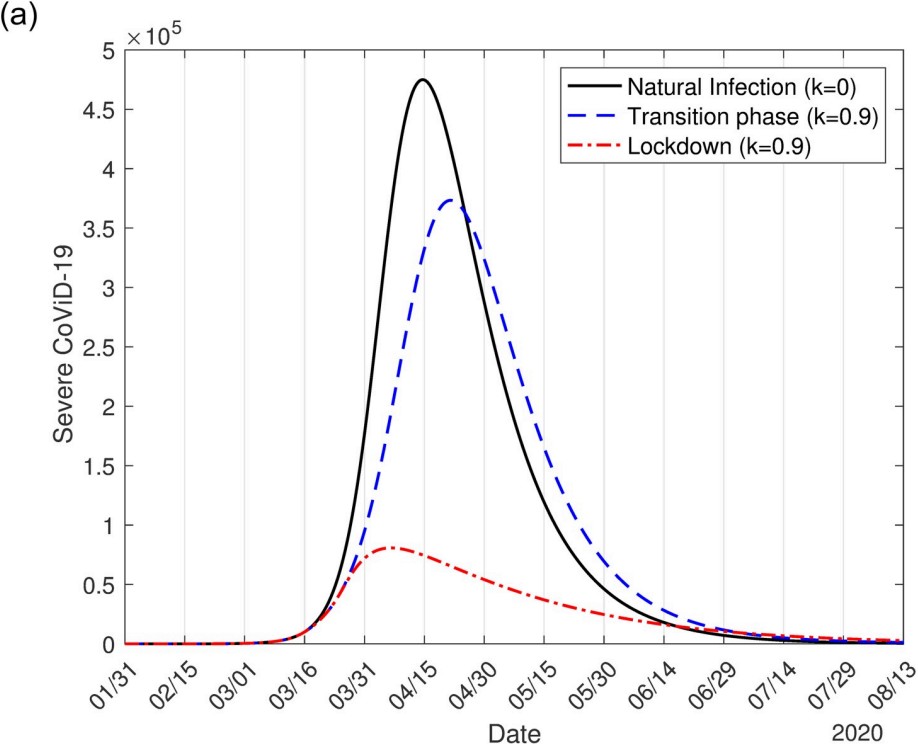

(b)

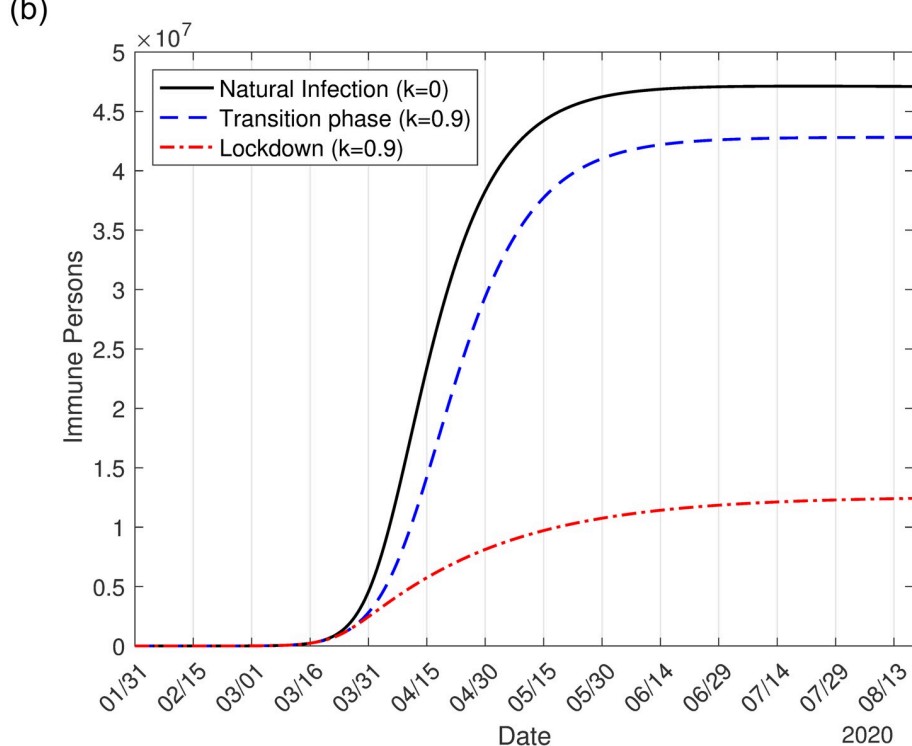

**Fig 5.** The curves of the natural epidemic ($k = 0$), the epidemic in the transition phase ($k = 0.9$), and the epidemic with lockdown ($k = 0.9$) (a), and the number of immune persons $I$ (b).

observe a decrease in the peaks of severe CoViD-19 $D_2$, which move to the right. Fig 5(a) shows the curves representing the natural epidemic, epidemic in transition, and epidemic in effective lockdown. In Fig 5(b), we show the number of immune persons $I$ corresponding to the three cases shown in Fig 5(a). The curves of $I$ follow a sigmoid shape.

In the natural epidemic, the numbers of immune persons $I_y$, $I_o$, and $I$ increase from zero to, respectively, 35 million, 12 million, and 47 million on June 15. In the epidemic with lockdown, the numbers are, on June 15, 8.37 million (24%), 3.09 million (25.75%), and 11.46 million (24.38%). Fig 5(b) shows only $I$ with and without interventions. The percentage between parentheses is the ratio between with and without interventions $I(k, \varepsilon)/I(0, 1)$ on June 15.

Let us compare the peak of $D_2$. The peaks for young, elder, and total persons in the natural epidemic are, respectively, 202, 600, 272, 700, and 475, 300, occurring on April 13, 15, and 14. In the epidemic with lockdown, the peaks for young, elder, and total persons are, respectively, 32, 850 (16%), 48, 190 (18%), and 80, 750 (17%), which occurred on April 5, 7, and 6. The percentage between parentheses is the ratio between natural epidemic and epidemic with lockdown $D_2(k, \varepsilon)/D_2(0, 1)$. The lockdown yielded a decrease to 17% compared to the natural epidemic.

Fig 6 shows the effective reproduction number $R_{ef}$ and $D_2$ during the epidemic in circulating (a) and lockdown (b) populations. To be fitted together in the same frame with $R_{ef}$, the curve of $D_2$ was divided by 1, 000 (a) and 12, 000 (b).

For the 90% of the population in lockdown in Spain, the basic reproduction number $R_0 = 8.0$ decreased to $R_{ef} = 0.771$ and $R_{ef} = 5.14$, respectively, in circulating and lockdown populations on March 16. During a short period of transition from natural to lockdown epidemic, the high effective reproduction number in the isolated population resulted in a high number of infections (see C.7(b) Fig in S1 Appendix, practically all cases are originated in lockdown population), which postpone the peak of the daily CoViD-19 cases to March 27. On March 24, when the effectiveness of lockdown is observed, another reduction in the effective reproduction number occurs to $R_{ef} = 0.382$ and $R_{ef} = 0.59$, respectively, in circulating and lockdown populations. Although $R_{ef} < 1$, the number of new cases of CoViD-19 does not decrease quickly due to the high number of susceptible individuals. Hence, Spain's example demonstrated that it is not enough to decrease the effective reproduction number below unity if the numbers of susceptible and infectious individuals are higher. On May 4 (phase 0 of release in Spain [21]) and June 8 (phase 3 of release) the effective reproduction number assumes, respectively, 0.53 and 0.51.

Fig 7 shows the curve $\Omega_d$ derived from $\Omega$ and the observed data in Spain (a), and the estimated curve of $\Omega$ with observed data, the curves $D_2$ and $\Omega_d$ with the observed data (b).

The peaks of the estimated curves $\Omega_d$ and $D_2$ are, respectively, 8, 922 and 80, 750, which occur on March 27 and April 6. The estimated daily cases' peak occurred one day later than the observed daily cases' peak, 9, 177 on March 26. However, for $k = 0.8$, the peaks of the estimated curves $\Omega_d$ and $D_2$ are, respectively, 9, 566 and 80, 020 occurring on March 28 and April 6. The value of the peak of the estimated daily cases and the date it occurred show that 90% of the population in lockdown explains better the daily observed data in Spain than 80%.

## Discussion

Firstly, we compare the epidemiological scenarios of the isolation in São Paulo State and the lockdown in Spain. Secondly, we address the critical question of the reliable estimation for $R_0$. Finally, we discuss the model proposed here.

(a)

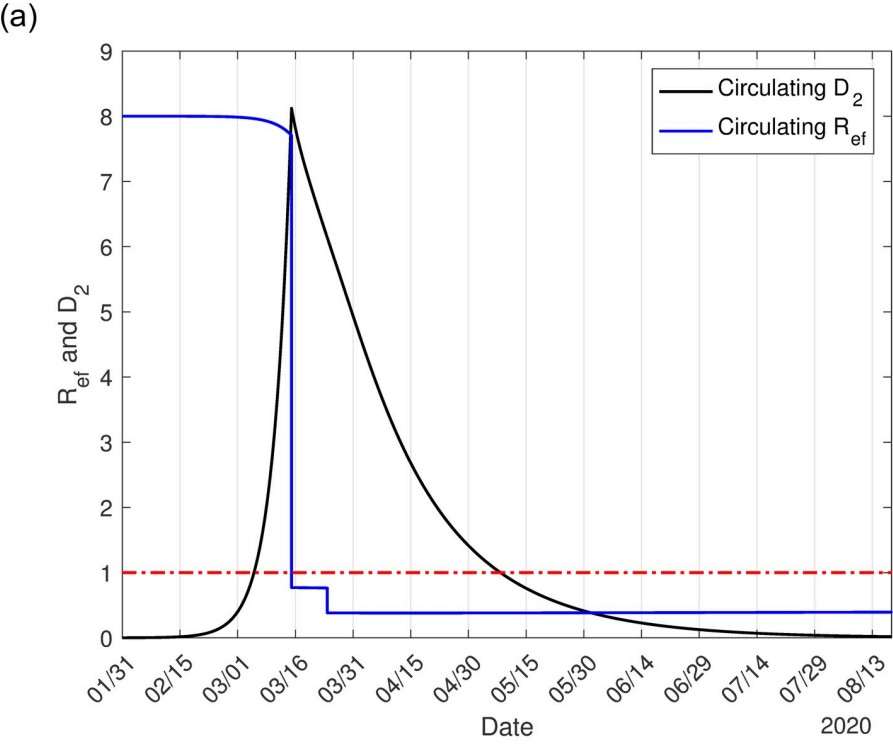

(b)

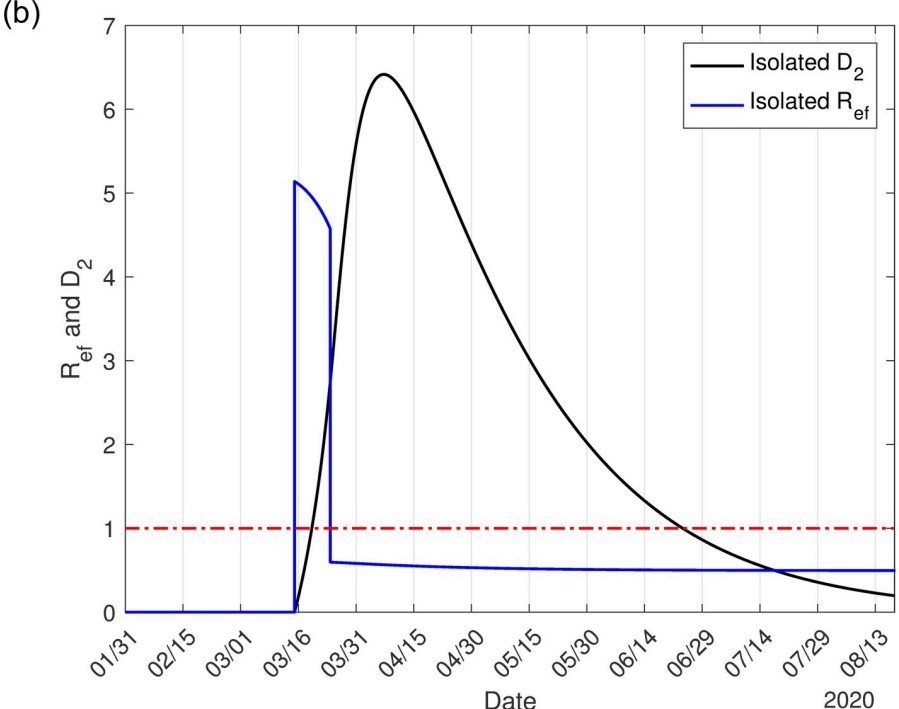

**Fig 6.** The effective reproduction number $R_{ef}$ for epidemic with lockdown in circulating (a) and locked-down (b) population in Spain. The number of severe covid-19 cases $D_2$ must be multiplied by $1,000$ (a) and $12,000$ (b).

(a)

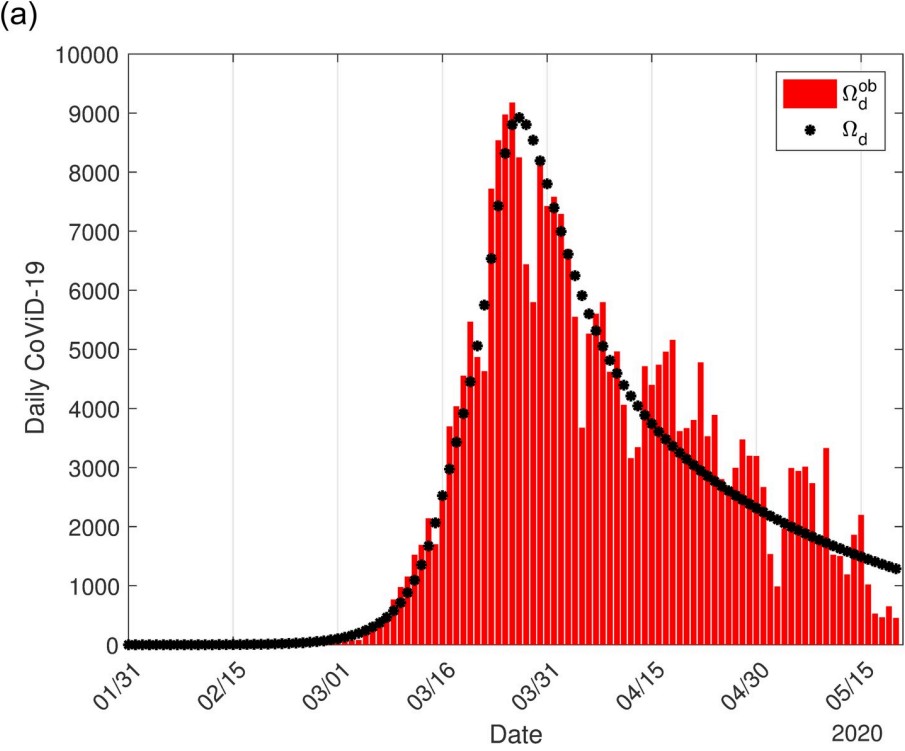

(b)

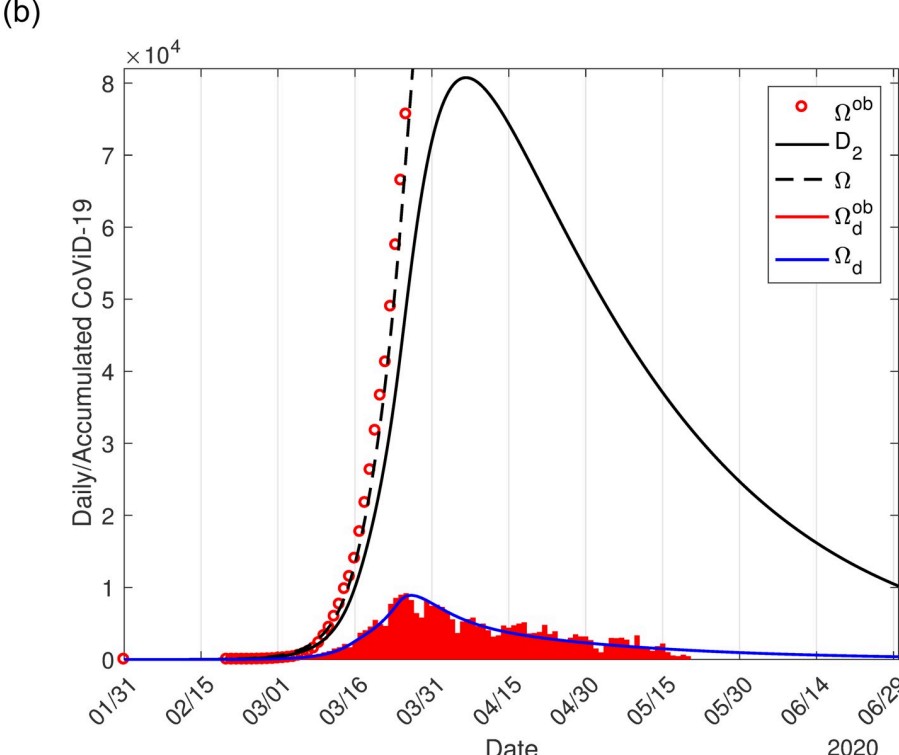

**Fig 7.** The calculated curve $\Omega_d$ and the observed daily cases in Spain (a), and the initial part of the estimated curve $\Omega$ with observed data $\Omega^{ob}$, the extended $\Omega_d$ and daily observed cases $\Omega_d^{ob}$, and severe covid-19 cases $D_2$ (b). All curves are the sum of the cases in circulating and locked-down populations.

## Comparing CoViD-19 epidemic in São Paulo State and Spain

To evaluate two different approaches to control the CoViD-19 epidemic, we considered two regions with similar population sizes—Spain (47.4 million, lockdown) and São Paulo State (44.6 million, partial quarantine). Spain has 6% more inhabitants than São Paulo State and implemented the lockdown 45 days after the first case of CoVid-19, 18 days later than the isolation implemented in São Paulo State after the first case. Spain has 48% less demographic density and 41% more proportion of elders than São Paulo State.

The widespread epidemic of CoViD-19 led Spain to implement lockdown, and the number of asymptomatic persons was higher as shown in Eq (C.2) in Section C of S1 Appendix, while in São Paulo State, the population was isolated earlier, for this reason, the number of asymptomatic persons was not so high as shown in Eq (C.1) in S1 Appendix. In the population in lockdown in Spain, the numbers of infectious and susceptible persons are, respectively, 3 and 1.8 times more than those found in isolation in São Paulo State. The number of new infection cases is proportional to the product of the numbers of infectious and susceptible persons; hence the population in lockdown in Spain has 5.4 times more risk of an outbreak of the epidemic than in São Paulo State. Thus, we neglected the SARS-CoV-2 transmission in the isolated population in São Paulo State; however, to explain Spain's observed data, we allowed a low transmission (through restricted contact occurring in the household and neighborhood) in the lockdown population. Indeed, the model provided that almost all severe CoViD-19 cases originated from the lockdown population in Spain, although $R_{ef}$ = 0.59. However, the epidemic of CoViD-19 in São Paulo State occurred in the circulating population, with the effective reproduction number jumping down to $R_{ef}$ = 4.35 when the isolation in 53% of the population was implemented on March 24, which decreased more to $R_{ef}$ = 2.1 with the adoption of protective measures on April 4. According to the definition in [7], São Paulo State is an example of mitigation, while Spain, of suppression.

The value of $R_{ef}$, from the beginning of the epidemic to the implementation of lockdown/isolation, decreased by 0.29 (with 8, 122 cases of severe CoViD-19) and 0.02 (with 397 cases), respectively, in Spain and São Paulo State. Moreover, the peak of severe CoViD-19 cases occurred on April 6 (66 days after the beginning of the epidemic), with 80, 750 cases in Spain, which is 17% of 474, 900 cases in the natural epidemic. In São Paulo State, the estimated peak of severe cases will occur on June 23 (118 days after the beginning of the epidemic) with 67, 140 cases (83% of Spain), 18% of 386, 400 cases in the natural epidemic. The 52 days gained by São Paulo State are precious to avoid overloading the health system, showing that the early adoption of isolation or lockdown is crucial to control the epidemic.

The estimated $R_0$ for CoViD-19 in Spain is 87% of that in São Paulo State, although the peak and the accumulated cases at the end of the first wave of the natural epidemic are 123% and 124% of those found in São Paulo State. We can understand this finding by analyzing the partial reproduction numbers $R_{0y}$ for young and $R_{0o}$ for elder subpopulations. For São Paulo State, we estimated $R_{0y}$ = 7.73 and $R_{0o}$ = 1.51. From the model, the fractions of young and elder susceptible persons reach, in the long-term epidemic, respectively, $s_y^{is*} = 1/R_{0y} = 0.13$ and $s_o^{is*} = 1/R_{0o} = 0.66$. However, when young and elder subpopulations are not separated but are interacting in the circulating population, we obtained $s_y^* = 0.1034$ and $s_o^* = 0.0017$. Notice that the difference $s_j^{is*} - s_j^*$, $j = y, o$, is the additional proportion of susceptible persons infected due to interaction, being 2.7% for young and 66% for elder persons, showing that elder persons are 24 times more risk than young persons when they interact. For Spain, we estimated $R_{0y}$ = 5.81 and $R_{0o}$ = 2.19, which are 75% and 145% of those estimated in São Paulo State. For the number of accumulated cases, Spain has 93% (young) and 179% (elder) of severe CoViD-19 cases of those found in São Paulo State. For the number of cases at the peak of the

epidemic, Spain has 90% (young) and 168% (elder) of those found in São Paulo State. Finally, in Spain, the most infections occurred in 90% of the lockdown population, with 25.8% being composed of the elder persons, while in São Paulo State, the infections were occurring in 47% of the circulating population with 15.3% being composed of the elder persons. Therefore, the higher number of cases with lower $R_0$ in Spain can be explained epidemiologically by lockdown, which allowed a higher number of elder persons in close contact with young persons, increasing the infection in the vulnerable elder subpopulation.

At the end of the first wave of the epidemic in quarantine, the accumulated severe CoViD-19 cases in Spain are 320, 200, and in São Paulo State, 386, 600 (121% of the cases in Spain), although the peak of the epidemic in Spain is higher. As a consequence, the lockdown implemented in Spain reduced the number of CoViD-19 cases (27% of the natural epidemic) more than isolation adopted in São Paulo State (41% of the natural epidemic), which impacts the number of immune (recovered) persons. The number of young and elder immune persons are, respectively, 15.14 million (99%) and 2.82 million (97%) for São Paulo State, and 9.13 million (99%) and 3.41 million (98%) for Spain. The percentage between parentheses is the ratio between the numbers of immune persons and new cases of CoViD-19 $I/\Phi$. In São Paulo State, the total number of immune persons at the end of the first wave of the epidemic is 40% of the population, while in Spain, 26.5%. On June 15 (beginning of release), the proportion of the São Paulo State's immune person is 16.7%. In Spain, on May 4 (phase 0 of release), June 8 (phase 3 of release), and 15, the proportions of the immune persons are, respectively, 18%, 23.5%, and 24.4%, showing that on June 8, Spain is close to the end of the first wave of the epidemic. However, if lockdown/isolation and protective measures are relaxed or abandoned, Spain and São Paulo State will be at higher risk to trigger a second wave of the epidemic due to an increased number of susceptible persons and a low number of immune persons at the end of the first wave of the epidemic.

At the end of the first wave of the epidemic in quarantine, the estimated number of deaths in Spain is 32, 150 (10% of all cases), while in São Paulo State is 23, 780 (6% of all cases), which is 74% of the total deaths in Spain. The São Paulo State's severity case fatality rate is 3.7 times higher in young and 1.4 times lower in the elder than in Spain (see Section C in S1 Appendix). Both severity cases and infection fatality rates for Spain for young and elder persons are around 30% and 138% of the São Paulo State rates. These rates in elder subpopulation could be explained by the life expectancy (São Paulo State has 78.4 *years* and Spain, 83.4 *years*), because 59.7% of deaths occurred in elder persons with 80 years old or more in Spain [20]. However, a higher proportion of uncontrolled comorbidity in the young subpopulation in São Paulo State increased the number of deaths [22].

The number of deaths was around 12% of severe CoViD-19 cases in Spain, while in São Paulo State, it was about 7% on May 20. The number of deaths is closely related to the number of severe CoViD-19 cases. As we pointed out, the close interaction between lockdown young and elder subpopulations (the presence of infectious young individuals increases the risk of infection among elders [23]) increased the epidemic in the elder subpopulation, increasing deaths. Moreover, the quick increase in the number of severe CoViD-19 in Spain overloaded hospitals and contributed to a rise in untreated patients' death, especially elders. The current relatively low number of fatalities in São Paulo State compared with that observed in Spain indicate that the health care system must be prepared to avoid hospital overload.

The daily registered CoViD-19 cases in B.1(a) and B.3(a) in Section B of S1 Appendix showed an increasing phase ($R_{ef} > 1$) followed by a decreasing phase ($R_{ef} < 1$) after reaching a maximum value at around $R_{ef} = 1$. The accumulated CoViD-19 cases shown in B.1(b) and B.3(b) Figs in S1 Appendix, however, showed a sigmoid shape curve, that is, a quick increase in the first phase ($R_{ef} > 1$, upward concavity) followed by a slow increase phase ($R_{ef} < 1$,

downward concavity) of the epidemic. When the daily cases attain their maximum value ($R_{ef} = 1$), the sigmoid curve changes the concavity, called the inflection point (or time). This shape can be used in connection with the efforts to eradicate an infection (vaccination or quarantine), which must be equal or greater than $f_{min} = 1 - 1/R_0$, the threshold of the fraction of susceptible individuals [10].

The curve of the accumulated CoViD-19 $\Omega$ and the observed data in Spain (C.7(b) and B.3 (b) Figs in S1 Appendix) showed the inflection point 11 days after the implementation of lockdown. It means that the lockdown resulted in $R_{ef} = 1$ soon. Letting $f_{min} = 0.7$, we must have $R_0 > 3.3$ to describe the observed epidemic after lockdown, but for $f_{min} = 0.8$, we must have $R_0 > 5$. However, the curve of $\Omega$ and observed data after the adoption of isolation in São Paulo State showed upward concavity after isolating 53% of individuals, and they reached the inflection point 80 days later (as we pointed out, $R_{ef} = 2.1$ on March 24.) Suppose that $R_{ef} = 1$ with 53%, hence, letting $f_{min} = 0.53$, we must have $R_0 > 2.1$ to describe the observed epidemic after isolation. However, the long period to reach the inflection point demonstrates that $R_0$ must be much higher than 2.1.

## Reliable estimation of $R_0$

Amer *et al.* [12] developed the IPR (Infected Patient Ratio) tool to measure the number of patients resulting from 1 primary infector during the incubation period. Using historical data from Italy, Germany, Spain, France, the United States of America, and China, they calculated a median of 16 patients infected by a primary infector during the incubation period. Accepting that the basic reproduction number $R_0$ is roughly associated with IPR, the estimated $R_0$ for São Paulo State (9.24) and Spain (8.0) are closer to the calculated IPR than estimations around 3. Moreover, Amer *et al.* observed in China that the average IPR dropped from 38 infected patients to 4 after only 12 days of lockdown (decreased by 89.5%). On the other hand, on March 28, 12 days after the lockdown in Spain, the effective reproduction number $R_{ef}$ was around 0.6, decreasing by 92.5%. Performing statistical analysis (segmented regressions), Santamaría and Hortal [13] described the effectiveness of lockdown in Spain associated with protective measures.

In the literature, the usually assumed basic reproduction number $R_0$ is around 2−3, see for instance [6, 7]. However, Li *et al.* [8] explicitly cited that, by using data from January 10 to February 8, 2020, they estimated the effective reproduction number $R_{ef}$, arguing that the most recent common ancestor could have occurred on November 17, 2019. The time elapsed from November 17, 2019 (the first case) to January 10, 2020 (the first day in the estimation) is 54 days. On January 23, 2020, Wuhan and other cities of Hubei province imposed a lockdown. As we pointed out in C.1(b) Fig in S1 Appendix, taking into account the entire data or restricting the interval of data around quarantine implementation, the estimated $R_{ef}$ must be lower.

From Figs 3 and 6, the effective reproduction number $R_{ef}$ for São Paulo State, 54 days after the beginning of the epidemic, is 2.1 (April 20), while for Spain, $R_{ef} = 0.6$ (March 25). In other words, using CoViD-19 data beginning from April 20 (São Paulo State) or March 25 (Spain), probably the estimated effective reproduction number will be close to those retrieved from Figs 3 and 6. On the other hand, if we estimate the basic reproduction number using the SIR model with different infective persons at $t = 0$, we obtained, using data collected from São Paulo State, $R_0 = 3.22$ (for $I(0) = 10$), or $R_0 = 2.66$ (for $I(0) = 25$), or $R_0 = 2.38$ (for $I(0) = 50$), with other initial conditions being $S(0) = 44.6$ million, and $R(0) = 0$. The SIR model is formulated considering only one class of infectious individuals. However, the available data at the beginning of the epidemic is the severe CoViD-19 cases, which are hospitalized and, probably,

they are not transmitting except to the hospital staff. Hence, the SEIR model is not appropriate to describe the CoViD-19 epidemic [9].

The reliable estimation of $R_0$ is essential because this number determines the magnitude of effort to eradicate infection. In the case of vaccination, the efforts to eradicate a disease must be vaccinating a fraction equal to or greater than $1 - 1/R_0$ of susceptible individuals [10]. In Yang [24], analyzing vaccination as a control mechanism, if $R_0$ is reduced by the vaccine to a value lower than one, the number of cases decreased following exponential-type decay, as we observed in Fig 6 describing the lockdown in Spain. Hence, instead of a vaccine, let us consider lockdown to control CoViD-19 transmission. If $R_0 = 3$, we must isolate at least 67% of the population, while for $R_0 = 8$, at least 87% of the population. As we have pointed out, 70% of the people in lockdown did not explain the CoViD-19 data in Spain, but 90% of the people in lockdown described better the observed data. Hence, our estimation of $R_0$ for Spain using the first period of CoViD-19 data is more reliable than that provided by the SEIR model.

In Yang *et al.* [11, 25], we estimated the additional mortality rates based on the observed data and concluded that their values fitted well at the beginning of the epidemic but did not provide reliable fitting in the long-term epidemic. For instance, that method of estimation pairing the numbers of new cases and deaths at the registering time resulted in deaths of 30% up to 80% of severe CoViD-19 cases at the end of the epidemic's first wave. For this reason, we had adopted a second method of estimation considering that the accumulated deaths in the elder subpopulation at the end of the first wave of the epidemic must be around 10%, underestimating the number of deaths at the beginning of the epidemic. Here, we improved the estimation of the additional mortality rates by pairing the numbers of new cases and deaths not at the registering time but delayed in 15 days, which is suggested by comparing the daily registered data of new cases with fatalities (see B.1 Fig in S1 Appendix). C.4 and C.8 Figs in S1 Appendix showed that this novel method of estimation fits relatively well during the three periods of the epidemic with $\Delta = 15$ *days*. However, we can vary $\Delta$ according to the period of the epidemic to obtain a better fitting. For instance, in C.8 Fig in S1 Appendix, the accumulated data of deaths corresponding to the natural epidemic is well fitted using $\Delta = 7$ *days*.

The concept of herd immunity is associated with the protection provided by immunized persons to a specific subpopulation under a higher risk of death caused by a syndrome or comorbidity. For instance, in the rubella infection, mass vaccination was planned to diminish the infection among pregnant women to reduce the number of congenital rubella syndrome [26]. The vaccination jumps down $R_{ef}$ as shown in Figs 3 and 6, and the isolation of a fraction $k$ of the susceptible persons. Different from the permanent reduction promoted by a vaccine, the herd protection implemented in a population reduces $R_{ef}$ temporarily and lasts whenever the population maintains adherence to lockdown/isolation and protective measures. Hence, the non-pharmaceutical interventions protect especially the elder subpopulation under higher risk of infection and death.

## Notes regarding the model

We formulated a deterministic compartment model to describe the quarantine as a control mechanism of the CoViD-19 epidemic. The model considered essential compartments according to the natural history of CoVid-19, and the depending on age fatality was incorporated considering two subpopulations. This model is minimalist also in retrieving the basic reproduction number $R_0$ analytically: Any addition in compartments and or age groups becomes this task extremely hard or impossible.

The model considered homogeneity in the spatial distribution, social contact and behavior, genetic, nutrition, etc. In Section A.4 in S1 Appendix, we present different mathematical

approaches to incorporate some elements of heterogeneity. The model did not take into account the loss of immunity neither the reinfection. SARS-CoV-2, like all RNA-based viruses, mutates faster and may originate variants of the original virus, which was not considered in the model.

We applied the model to describe the data recorded from São Paulo State and Spain. The vital dynamic and natural history of CoViD-19 parameters' values in Table 2 can be evaluated for other countries or regions. Using these values, the SARS-CoV-2 transmission and intervention parameters can be fitted against the observed CoViD-19 data. The further observed severe CoViD-19 cases and deaths must be confronted with the model's predictions while the interventions last. Once the model's predictability was verified, the epidemiological scenario of the isolation can be considered as the background to evaluate (or predict the outcomes of) the relaxation strategies.

## Conclusion

In the absence of effective treatment and vaccine, the lockdown adoption at the very beginning of the epidemic is recommended to control the SARS-CoV-2 with high transmissibility and lethality. The second strategy is the implementation of isolating, as São Paulo State did. In this strategy, the epidemic curve of CoViD-19 in the circulating population is flattened to avoid the overloading in hospitals, and the immunization by the natural epidemic is increased—unfortunately, the number of deaths due to CoViD-19 increases. The third strategy, the adoption of lockdown, is recommended when the epidemic is out of control, and Spain is an example. In the second and third strategies to control the CoViD-19 epidemic, the severe CoViD-19 data before the adoption of isolation or lockdown are used to estimate the basic reproduction number.

Quarantine (isolation and lockdown) is a valuable measure to control an epidemic with high lethality. Due to the health care system's critical situation, Spain imposed a rigid quarantine (lockdown), which impacted the fast ascending phase of the epidemic by reducing $R_{ef}$ below one. This reduction resulted in a peak of 80, 750 cases occurring 20 days after the implementation of lockdown (on March 16, $\Omega^{ob}$ = 14, 011). Convinced by the epidemiological situation in Spain, São Paulo State implemented partial quarantine (isolation) and, as a result, the peak of 67, 140 cases will occur 91 days after the isolation (on March 24, $\Omega^{ob}$ = 810). The relatively early implementation of the isolation in São Paulo State somehow avoided the overloading in the health care system by flattening the epidemic curve.

The proportion of the immune (recovered) persons at the end of the first wave of epidemic is 40% of São Paulo State's population and 26% of Spain's population. On June 8 (phase 3 of the release in Spain [21]), the effective reproduction number was 0.51, and 23.5% of the population was immune, showing that Spain was close to the end of the first wave of the epidemic. This relatively safe epidemiological scenario was favorable to implement a carefully planned relaxation (release). However, on June 15, the effective reproduction number was 0.98, but in the ascending phase of the epidemic, and 16.7% of the population was immune, showing that São Paulo State was far from the end of the epidemic. The release of the isolated persons in this unfavorable epidemiological scenario may enhance the already intense transmission of SARS-CoV-2. Additionally, the abandonment of protective measures (face mask, washing hands, and social distancing) may result in a fierce retaken of the epidemic, especially in Spain.

## Supporting information

**S1 Appendix.**
(PDF)

## Acknowledgments

We thank to anonymous referees for providing comments and suggestions, which contributed to improving this paper.

## Author Contributions

**Conceptualization:** Hyun Mo Yang.

**Data curation:** Luis Pedro Lombardi Junior.

**Formal analysis:** Hyun Mo Yang.

**Investigation:** Hyun Mo Yang, Luis Pedro Lombardi Junior, Fábio Fernandes Morato Castro, Ariana Campos Yang.

**Methodology:** Hyun Mo Yang, Ariana Campos Yang.

**Software:** Luis Pedro Lombardi Junior.

**Supervision:** Hyun Mo Yang.

**Validation:** Fábio Fernandes Morato Castro, Ariana Campos Yang.

**Visualization:** Hyun Mo Yang, Luis Pedro Lombardi Junior.

**Writing – original draft:** Hyun Mo Yang.

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
