## [Decision Letter · Decision Letter 0]

30 Sep 2020

PONE-D-20-19611

Mathematical modeling of the transmission of SARS-CoV-2 - Evaluating the impact of isolation in São Paulo State (Brazil) and lockdown in Spain associated with protective measures on the epidemic of covid-19

PLOS ONE

Dear Dr. Yang,

Thank you for submitting your manuscript to PLOS ONE. After careful consideration, we have decided that your manuscript does not meet our criteria for publication and must therefore be rejected.

Specifically:

Your manuscript was reviewed by one expert in the field who found many important technical issues in your submission and produced a very strong recommendation. The most concerning are comments on validity of your model and estimates of R0. Please carefully consider the attached comments which we hope will help you to improve your work.

I am sorry that we cannot be more positive on this occasion, but hope that you appreciate the reasons for this decision.

Yours sincerely,

Yury E Khudyakov, PhD

Academic Editor

PLOS ONE

Reviewers' comments:

Reviewer's Responses to Questions

**Comments to the Author**

1. Is the manuscript technically sound, and do the data support the conclusions?

Reviewer #1: No

2. Has the statistical analysis been performed appropriately and rigorously? 

Reviewer #1: No

3. Have the authors made all data underlying the findings in their manuscript fully available?

Reviewer #1: No

4. Is the manuscript presented in an intelligible fashion and written in standard English?

Reviewer #1: No

5. Review Comments to the Author

Reviewer #1: 1) It is too long. Could you compress the paper ?

2) The estimated R0 seems to very large. The author(s) should carefully discuss it. At least, I understand that the author(s) did something wrong. I am wondering that it is reasonable to evaluate R0 at steady state. It might affect the severe upper bias in R0. At least, they have to discuss carefully about other estimated R0 in the same data. Or they have to provide the estimation results in other area and compare them. If the estimation result in other area by the author(s) will be consistently higher than the earlier studies, the author(s) methodology should be concluded to be bias in R0. These robustness check should be necessary.

3) We need to confidence interval of the estimated parameter. If CI of R0 include two, it might not be statistically difference among the earlier study.

4) I understand that the model in this paper was deterministic SIR model. If so, you mention it in Introduction and never mention about individual based model, which is not related this paper.

5) If you adopt deterministic SIR model without CI, you have to explain carefully the reason why　do they adopt it. I think that it is not meaningless the estimation without CI.

6) The estimation of R0 should be done using epidemic curve, which indicates number of patient at onset date. the author(s) might use the data of the number of reported patients in each day. If so, the authors should take delay to report from onset into the model.

7) The author(s) assume rubella is airborne infection, but I do not agree with it. Measles or varicella are airborne infection.

8) Is really protection measure with mask or handwashing reduce R ? I think that reduction in going out before lock down seems to be more important proved by the earlier paper.

9) Why the author(s) ignore less susceptible in children though the model has birth rate ? I think less susceptible in children is the most important question to answer for control of COVID-19.

10) I do not agree with high lethality in COVID-19 to achieve herd immunity. Because the severe patient do not contact anyone other than hospila staff and thus almost not infectiousness. If the author(s) do not think so, they have to provide careful discussion.

11) How do you the effect of lock down ? Can SIR model express lock down even it is complete homogenous model ?

minor points

1) The author(s) use covid-19 instead of COVID-19.

2) The author(s) should change new to novel.

3) What mean "immune person" ?

6. PLOS authors have the option to publish the peer review history of their article (what does this mean?). If published, this will include your full peer review and any attached files.

Reviewer #1: No

- - - - -

---

## [Decision Letter · Decision Letter 1]

29 Dec 2020

PONE-D-20-19611R1

Mathematical modeling of the transmission of SARS-CoV-2 - Evaluating the impact of isolation in São Paulo State (Brazil) and lockdown in Spain associated with protective measures on the epidemic of covid-19

PLOS ONE

Dear Dr. Yang,

Thank you for submitting your manuscript to PLOS ONE. After careful consideration, we feel that it has merit but does not fully meet PLOS ONE’s publication criteria as it currently stands. Therefore, we invite you to submit a revised version of the manuscript that addresses the points raised during the review process.

In particular, please respond to Reviewer 2's comment on your estimate of R0 = 8 being too large for COVID-19. Please also prepare a table of estimated parameters (with confidence intervals, if possible) of your model, in response to Reviewer 3's comment.

We look forward to receiving your revised manuscript.

Kind regards,

Siew Ann Cheong, Ph.D.

Fernanda C. Dórea

Academic Editors

PLOS ONE

Journal Requirements:

2. We note that your manuscript is not formatted using one of PLOS ONE’s accepted file types. Please reattach your manuscript as one of the following file types: .doc, .docx, .rtf, or .tex (accompanied by a .pdf).

If your submission was prepared in LaTex, please submit your manuscript file in PDF format and attach your .tex file as “other.”

3. Thank you for updating your data availability statement. You note that your data are available within the Supporting Information files, but no such files have been included with your submission. At this time we ask that you please upload your minimal data set as a Supporting Information file, or to a public repository such as Figshare or Dryad.

Please also ensure that when you upload your file you include separate captions for your supplementary files at the end of your manuscript.

As soon as you confirm the location of the data underlying your findings, we will be able to proceed with the review of your submission.

Additional Editor Comments (if provided):

Reviewers' comments:

Reviewer's Responses to Questions

**Comments to the Author**

1. If the authors have adequately addressed your comments raised in a previous round of review and you feel that this manuscript is now acceptable for publication, you may indicate that here to bypass the “Comments to the Author” section, enter your conflict of interest statement in the “Confidential to Editor” section, and submit your "Accept" recommendation.

Reviewer #2: All comments have been addressed

Reviewer #3: (No Response)

2. Is the manuscript technically sound, and do the data support the conclusions?

Reviewer #2: Yes

Reviewer #3: Partly

3. Has the statistical analysis been performed appropriately and rigorously? 

Reviewer #2: Yes

Reviewer #3: N/A

4. Have the authors made all data underlying the findings in their manuscript fully available?

Reviewer #2: Yes

Reviewer #3: Yes

5. Is the manuscript presented in an intelligible fashion and written in standard English?

Reviewer #2: Yes

Reviewer #3: Yes

6. Review Comments to the Author

Reviewer #2: Thank you for the opportunity in consideration of reviewing the given manuscript. The manuscript presenting the epidemic modelling of COVID-19 among individual state in Brail and Spain. The study outcomes providing promising view in understanding of mitigation measures in these two places. However, I want to make notice to authors some minor comments

Materials and methods section should after introduction

In Italy and Spain, because of continuous lockdown and self-isolation it was possible to control the pandemic severity. In Introduction I found an orphan statement “S~ao Paulo State, however, implemented a partial quarantine (isolation) in the population to avoid critical epidemiological scenarios that occurred in Spain and Italy” [1][2]. To in line with this statement authors are recommended to include a citation of recent studies on these nations such as

1. Antonio Guirao, The Covid-19 outbreak in Spain. A simple dynamics model, some lessons, and a theoretical framework for control response, Infectious Disease Modelling, Volume 5, 2020, Pages 652-669, https://doi.org/10.1016/j.idm.2020.08.010.

2. Battineni, G., Chintalapudi, N. and Amenta, F. (2020), "SARS-CoV-2 epidemic calculation in Italy by SEIR compartmental models", Applied Computing and Informatics, https://doi.org/10.1108/ACI-09-2020-0060

From the model parameters, we obtained for the basic reproduction number R0 = 9:24 for S~ao Paulo State, and R0 = 8 for Spain.

The R0 seems very high, could you check the bias in this finding? With this rate, almost everyone in the country has to be infected. The high hit epidemic place like Lombardy, Italy R0 has registered about 6.5. I am surprising that this is the first study that met COVID-19 R0≥8. I would like to request authors for cross checking. However, this sentence should be in results section.

Please mention the limitation of the proposed model at bottom of discussion part

Reviewer #3: The paper deals with the issues on COVID-19 in Sao Paolo state and Spain. I have some comments on this.

1. What is the rationale behind choosing one state in Brazil and Spain? Is there any similarity between these two regions (one state and one country) for which they might be comparable with respect to some features of COVID outbreak?

2. The model is too complicated involving too many parameters. Parameters of such a model are very difficult to estimate, unless one has very accurate data on all compartments and also about the transition between two compartments. The authors should clarify this to increase its readability. Moreover, this model depends on too many parameters; a rigorous sensitivity analysis is required to study the effect of variation in the parameters.

3. There are many parameters whose estimates are not available (indicated by ***). Without any clear idea the analysis would be incomplete. Also the authors should study the effect of these parameters.

4. The paper is too large. A shortened version is welcome.

I suggest a major revision in view of the above comments before reaching any decision regarding acceptance or rejection of this manuscript.

7. PLOS authors have the option to publish the peer review history of their article (what does this mean?). If published, this will include your full peer review and any attached files.

Reviewer #2: No

Reviewer #3: No

---

## [Author Response · Author response to Decision Letter 1]

11 Jan 2021

We uploaded three files: ResponsetoEditors.pdf, ResponsetoReferee2.pdf and ResponsetoReferee3.pdf.

---

## [Decision Letter · Decision Letter 2]

11 Mar 2021

PONE-D-20-19611R2

Mathematical modeling of the transmission of SARS-CoV-2 - Evaluating the impact of isolation in São Paulo State (Brazil) and lockdown in Spain associated with protective measures on the epidemic of covid-19

PLOS ONE

Dear Dr. Yang,

Thank you for submitting your manuscript to PLOS ONE. After careful consideration, we feel that it has merit but does not fully meet PLOS ONE’s publication criteria as it currently stands. Therefore, we invite you to submit a revised version of the manuscript that addresses the points raised during the review process.

In particular, please address all the concerns of Reviewer 3, on the estimation of model parameters, and either estimate the covariances between parameters, or justify why they should be treated as zero.

We look forward to receiving your revised manuscript.

Kind regards,

Siew Ann Cheong, Ph.D.

Academic Editor

PLOS ONE

Journal Requirements:

Reviewers' comments:

Reviewer's Responses to Questions

**Comments to the Author**

1. If the authors have adequately addressed your comments raised in a previous round of review and you feel that this manuscript is now acceptable for publication, you may indicate that here to bypass the “Comments to the Author” section, enter your conflict of interest statement in the “Confidential to Editor” section, and submit your "Accept" recommendation.

Reviewer #2: All comments have been addressed

Reviewer #3: (No Response)

2. Is the manuscript technically sound, and do the data support the conclusions?

Reviewer #2: Yes

Reviewer #3: Partly

3. Has the statistical analysis been performed appropriately and rigorously? 

Reviewer #2: Yes

Reviewer #3: No

4. Have the authors made all data underlying the findings in their manuscript fully available?

Reviewer #2: Yes

Reviewer #3: Yes

5. Is the manuscript presented in an intelligible fashion and written in standard English?

Reviewer #2: Yes

Reviewer #3: Yes

6. Review Comments to the Author

Reviewer #2: Author sufficiantly addressed the raised comments. It is now ready to accept in the current format

Reviewer #3: The authors tried to address the issues raised by the reviewers. However, to my understanding, not all issues are addressed directly and adequately. During the sensitivity analysis, they have assumed the covariances to be zero without explaining the rationale behind it. If the concept of considering covariances occurs, it indicates that the parameters are treated as random but in the entire analysis (mathematical and numerical) all parameters are treated as constants. Moreover the reason of considering one state in a South American country and one country in Europe is not clear. One could have considered many such pairs. There are also some unambiguity in parameter estimation especially R0. Moreover, although there is a sense of heterogeneity on the model, the authors did not explicitly (and mathematically) use this idea in the model evaluation and analysis.

Based on my understanding, I recommend rejecting the paper.

7. PLOS authors have the option to publish the peer review history of their article (what does this mean?). If published, this will include your full peer review and any attached files.

Reviewer #2: **Yes: **Dr. Gopi Battineni

Reviewer #3: No

---

## [Author Response · Author response to Decision Letter 2]

14 Mar 2021

Please, see editor.pdf, ResponsetoReviewer2.pdf and ResponsetoReviewer3.pdf.

---

## [Decision Letter · Decision Letter 3]

27 Apr 2021

PONE-D-20-19611R3

Mathematical modeling of the transmission of SARS-CoV-2 - Evaluating the impact of isolation in São Paulo State (Brazil) and lockdown in Spain associated with protective measures on the epidemic of covid-19

PLOS ONE

Dear Dr. Yang,

Thank you for submitting your manuscript to PLOS ONE. After careful consideration, we feel that it has merit but does not fully meet PLOS ONE’s publication criteria as it currently stands. Therefore, we invite you to submit a revised version of the manuscript that addresses the points raised during the review process.

Please address the comments provided by Reviewer 4. I expect that no further review will be necessary.

We look forward to receiving your revised manuscript.

Kind regards,

Siew Ann Cheong, Ph.D.

Academic Editor

PLOS ONE

Journal Requirements:

Reviewers' comments:

Reviewer's Responses to Questions

**Comments to the Author**

1. If the authors have adequately addressed your comments raised in a previous round of review and you feel that this manuscript is now acceptable for publication, you may indicate that here to bypass the “Comments to the Author” section, enter your conflict of interest statement in the “Confidential to Editor” section, and submit your "Accept" recommendation.

Reviewer #4: (No Response)

Reviewer #5: (No Response)

2. Is the manuscript technically sound, and do the data support the conclusions?

Reviewer #4: (No Response)

Reviewer #5: Yes

3. Has the statistical analysis been performed appropriately and rigorously? 

Reviewer #4: (No Response)

Reviewer #5: Yes

4. Have the authors made all data underlying the findings in their manuscript fully available?

Reviewer #4: (No Response)

Reviewer #5: Yes

5. Is the manuscript presented in an intelligible fashion and written in standard English?

Reviewer #4: (No Response)

Reviewer #5: Yes

6. Review Comments to the Author

Reviewer #4: (No Response)

Reviewer #5: The manuscript is well written and the mathematical model is appreciable. Authors expression of English language too have improved.

7. PLOS authors have the option to publish the peer review history of their article (what does this mean?). If published, this will include your full peer review and any attached files.

Reviewer #4: **Yes: **Faten Amer

Reviewer #5: **Yes: **Bamidele Tolulope Odumosu

---

## [Editor Report · Decision Letter 4]

14 May 2021

Mathematical modeling of the transmission of SARS-CoV-2 - Evaluating the impact of isolation in São Paulo State (Brazil) and lockdown in Spain associated with protective measures on the epidemic of covid-19

PONE-D-20-19611R4

Dear Dr. Yang,

We’re pleased to inform you that your manuscript has been judged scientifically suitable for publication and will be formally accepted for publication once it meets all outstanding technical requirements.

Kind regards,

Siew Ann Cheong, Ph.D.

Academic Editor

PLOS ONE
---

## [Editor Report · Acceptance letter]

2 Jun 2021

PONE-D-20-19611R4 

Mathematical modeling of the transmission of SARS-CoV-2 – Evaluating the impact of isolation in São Paulo State (Brazil) and lockdown in Spain associated with protective measures on the epidemic of CoViD-19 

Dear Dr. Yang:

I'm pleased to inform you that your manuscript has been deemed suitable for publication in PLOS ONE. Congratulations! Your manuscript is now with our production department. 

Kind regards, 

on behalf of

Dr. Siew Ann Cheong 

Academic Editor

PLOS ONE